# Uncovering Semantic Selectivity of Latent Groups in Higher Visual Cortex with Mutual Information-Guided Diffusion

**Yule Wang, Joseph Yu, Chengrui Li, Weihan Li, Anqi Wu**
Georgia Institute of Technology
Atlanta, GA 30332, USA
{yulewang, jyu425, cnlichengrui, weihanli, anqiwu}@gatech.edu

## Abstract

Understanding how neural populations in higher visual areas encode object-centered visual information remains a key challenge in computational neuroscience. Prior work has examined representational alignment between artificial neural networks and the visual cortex, but such findings are indirect and provide limited insight into the structure of the neural population coding. Decoding-based methods can recover semantic features from neural activity, yet they do not reveal how these features are organized. This leaves an open question: *how feature-specific visual information is distributed across neural populations, and whether it forms structured, semantically meaningful subspaces.* To address this, we introduce MIG-Vis, a method that leverages diffusion models to visualize and validate visual-semantic attributes encoded in neural latent subspaces. MIG-Vis first learns a group-wise disentangled neural latent representation using a variational autoencoder. It then uses mutual information (MI)–guided diffusion synthesis to visualize the visual-semantic features encoded in each latent group. We validate MIG-Vis on multi-session neural spiking datasets from the inferior temporal (IT) cortex of two macaques. The synthesized results show that MIG-Vis identifies neural latent groups with clear semantic selectivity to diverse visual features, including object pose, inter-category variation, and intra-category content. These findings provide direct and interpretable evidence of structured semantic representation in higher visual cortex. The code repository of MIG-Vis is available at: https://github.com/BRAINML-GT/MIG-Vis.

## 1 Introduction

Determining how populations of neurons in higher visual areas represent object-centred visual information remains a central question in computational neuroscience (DiCarlo et al., 2012; Yamins & DiCarlo, 2016). A major line of research (Lindsey & Issa, 2024; Xie et al., 2024) has explored representational alignment between deep neural networks (DNNs) and primate visual cortex, showing that DNNs trained for object recognition—especially those with disentangled representation subspaces—closely resemble inferior temporal (IT) cortex activity. Nevertheless, these findings are indirect, relying on artificial model architecture and specially designed representational similarity metrics. Meanwhile, previous works have focused on single-unit selectivity (Rust & DiCarlo, 2010) or decoding-based methods (Freiwald & Tsao, 2010; Chang & Tsao, 2017) that quantify semantic features like object category or viewpoint in higher visual cortex. Earlier studies have also used pre-trained diffusion models with fMRI data (Luo et al., 2023a;b; Cerdas et al., 2024) to verify and classify that various brain regions specialize in processing certain categories' information.

However, existing approaches do not extract semantically interpretable neural representations from electrophysiological recordings in higher visual cortex. No study has mapped the organization and structural patterns of higher visual area neural populations to distinct visual attributes. In practice, addressing this scientific problem is challenging due to single-unit neural activities in the higher visual cortex exhibit mixed selectivity to multiple visual-semantic features (Chang & Tsao, 2017). We also verify this finding through an empirical study on the IT cortex of macaques in a passive object recognition task (Majaj et al., 2015) (the results are presented in Fig. 1).

Figure 1: **(A)** Single-neuron decoding results show that IT neurons exhibit mixed selectivity, contributing to both low-level pose attributes (e.g., rotation) and high-level semantic features (e.g., category identity) to different extents (indicated by line thickness). **(B)** Example images varying along each visual-semantic feature.

To address this gap, we propose **MIG-Vis** (**M**utual **I**nformation-**G**uided Diffusion for uncovering semantic selectivity of neural latent groups in higher **Vis**ual cortex), a method that identifies interpretable neural latent subspaces and visualizes the semantic meaning encoded in each subspace. We assume that multiple latent dimensions form a group that encodes a specific type of semantic feature. For example, object category may be represented by one latent group, while rotation is represented by another. To capture this structure, we use a **group-wise disentangled** variational autoencoder (Li et al., 2025), which learns latent groups composed of multiple dimensions, each corresponding to a distinct semantic feature type. Within each group, individual dimensions can still capture different aspects of that feature type, such as shape, texture, or lighting within a category-related group.

Given the learned neural latent space, our next goal is to understand the visual-semantic features that a certain latent group encodes. We achieve this by perturbing the latent and generating corresponding images, then comparing them to the original image to observe the semantic changes introduced by the perturbation. Traditionally, this can be done using a neural-to-image decoder to map a perturbed neural latent to an image (Whiteway et al., 2021). However, such a decoder tends to produce a single best reconstruction and may smooth out subtle variations in the latent space. As a result, small perturbations may not lead to clearly distinguishable semantic changes. An alternative is to use diffusion with guidance. Prior work typically guides diffusion by optimizing simple statistical moments of target neural dimensions, such as absolute activation values (Luo et al., 2023a) or variance (Wang et al., 2024). Such approaches are especially common in the fMRI literature, where neural activity is directly manipulated: zero indicates no activation, and larger positive values correspond to stronger responses. In our case, the neural representation lies in a learned latent space where both positive and negative values carry meaningful but distinct semantics. Thus, maximizing magnitude or variance does not result in meaningful semantic change encoded in neural latent space.

To address limitations in existing methods, we perturb the latent representation by adding or subtracting values along specific dimensions and seek to generate images that reflect these perturbed latents. Rather than relying on a decoder or diffusion guided by magnitude or variance maximization, we steer the diffusion process by maximizing the mutual information (MI) (Hjelm et al., 2018) between the synthesized image and the perturbed latent. MI captures the full statistical dependence between them, encouraging the generated image to retain the semantic changes introduced by the perturbation. Compared to direct neural-to-image decoding, MI-guided diffusion reduces the risk of collapsing semantic variations into an averaged reconstruction and is better suited to reveal how visual meaning changes along different latent directions.

We evaluate MIG-Vis on multi-session neural spiking datasets (Majaj et al., 2015) from the IT cortex of two macaques performing a passive object recognition task. Diffusion-based synthesis shows that MIG-Vis learns disentangled neural latent groups with distinct visual-semantic selectivity. By generating images guided by maximizing mutual information between the output and each latent group, we identify clear semantic roles across groups, including pose, inter-category variation, and intra-category content details. We further apply this interpretation method within each group to reveal fine-grained semantic structure.

## 2 PRELIMINARIES

**Problem Formulation.** For each single trial, data is composed of neural population and stimulus image pairs: $(\mathbf{x}, \mathbf{y})$. The neural population is denoted as $\mathbf{x} \in \mathbb{R}^N$, where $N$ is the number of recorded neural units. The corresponding image is represented as $\mathbf{y} \in \mathbb{R}^{1 \times H \times W}$, where $H$ and $W$

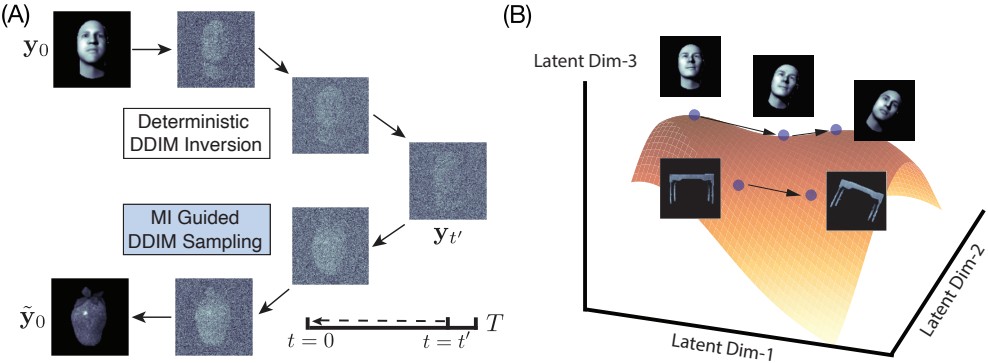

Figure 2: (A) **The semantic image editing procedure.** This consists of a deterministic forward process from $t = 0$ to an intermediate timestep $t = t'$, followed by a neural-guided deterministic synthesis process back to $t = 0$. (B) **Perturbation process.** We show the neural manifold of the latent group $\mathbf{z}_1$, which encodes object pose. We traverse along its first group. Starting from the original image, we slightly adjust the latent variable to obtain perturbed latents and generate the corresponding images by maximizing mutual information with these perturbed representations. Since the perturbations are applied within the pose subspace, the resulting image changes consistently reflect rotation variations.

denote the height and width of the gray-scale image. We develop a group-wise disentangled VAE based on Li et al. (2025) to infer the neural latent groups, denoted as $\mathbf{z} = [\mathbf{z}_1, \dots, \mathbf{z}_G]^\top \in \mathbb{R}^D$, in which $D$ is the latent dimension number and $G$ is the number of groups. Each latent group is set to have the same dimensionality $d_g$, thus $D = G \times d_g$. Our ultimate goal is to investigate the specific visual-semantic encoding of the $g$-th neural latent group.

**Classifier-Guided Diffusion Models.** Given a set of image data samples $\mathbf{y}$, a diffusion probabilistic model (Ho et al., 2020; Wang et al., 2023) estimates its density $p_{\text{data}}(\mathbf{y})$ by first perturbing the data points in a $T$-step forward process: $q\left(\mathbf{y}_t \mid \mathbf{y}_0\right) := \mathcal{N}\left(\mathbf{y}_t; \sqrt{\alpha_t}\mathbf{y}_0, (1 - \alpha_t)\,\mathbf{I}\right)$, where $\{\alpha_t\}_{t=1}^T$ denotes a noise-schedule. In the reverse process, a neural network $\boldsymbol{\epsilon}_{\boldsymbol{\theta}}\left(\mathbf{y}_t, t\right)$ parameterized by $\boldsymbol{\theta}$ is trained to estimate the introduced noise at each timestep $t$. On the other hand, classifier-guided diffusion models (Dhariwal & Nichol, 2021) enable conditional generation $p(\mathbf{y} \mid \mathbf{z})$ given a class label $\mathbf{z}$. Formally, given the class label $\mathbf{z}$ and a guidance scale $\eta > 0$, by Bayes' rule, the conditional distribution of sampled images $\mathbf{y}$ would be: $p^\eta(\mathbf{y} \mid \mathbf{z}) \propto p(\mathbf{y})p(\mathbf{z} \mid \mathbf{y})^\eta$. By taking the log-derivative on both sides, this gives:

$$\nabla_{\mathbf{y}_t} \log p^\eta\left(\mathbf{y}_t \mid \mathbf{z}\right) = \underbrace{\nabla_{\mathbf{y}_t} \log p_{\boldsymbol{\theta}}(\mathbf{y}_t)}_{\text{est. by } \boldsymbol{\epsilon}_\theta(\mathbf{y}_t, t)} + \eta \underbrace{\nabla_{\mathbf{y}_t} \log p_{\boldsymbol{\phi}}\left(\mathbf{z} \mid \mathbf{y}_t\right)}_{\text{est. by guidance}}. \tag{1}$$

The unconditional score term on the right-hand side (RHS) is a scaled form of the predicted noise, i.e., $\nabla_{\mathbf{y}_t} \log p(\mathbf{y}_t) = -\boldsymbol{\epsilon}_\theta(\mathbf{y}_t, t)/\sqrt{1 - \alpha_t}$. We propose a novel MI-based classifier parameterized by $\boldsymbol{\phi}$ to estimate $p\left(\mathbf{z} \mid \mathbf{y}_t\right)$, and the second term on the RHS corresponds to its log-derivative.

## 3 METHODOLOGY

In the following, we first describe how MIG-Vis infers a neural latent space via a group-wise disentangled VAE. We then present our approach for visualizing and interpreting visual-semantic variations encoded by each latent group through mutual information-guided diffusion synthesis.

### 3.1 INFERRING GROUP-WISE DISENTANGLED NEURAL LATENT SUBSPACE

To uncover interpretable visual–semantic structure in neural populations, we use a group-wise disentangled VAE that learns latent subspaces corresponding to different semantic factors. Traditional disentangled VAEs (Higgins et al., 2017; Chen et al., 2018) assume that each semantic factor is represented by a single independent latent dimension. However, this assumption is restrictive for high-level visual attributes, such as object category or 3D rotation, which typically require multiple latent dimensions to be represented adequately.

Hence, we relax the single-dimension assumption by using a group-wise disentangled VAE (Esmaeili et al., 2019; Li et al., 2025), which encourages statistical independence between multi-dimensional neural latent groups. As suggested by previous works (Locatello et al., 2019), well-disentangled semantic latents seemingly cannot be inferred without (implicit) supervision. We incorporate certain image attributes (i.e., rotation angles and category identity) as supervision to inform the latent subspace learning. The supervision labels are concatenated into a vector $\mathbf{u} \in \mathbb{R}^M$, where $M$ denotes its dimensionality. We decompose the neural latent vector $\mathbf{z}$ into supervised latent groups $\mathbf{z}^{(s)}$ and unsupervised latent groups $\mathbf{z}^{(u)}$, such that $\mathbf{z} = \left[\mathbf{z}^{(s)}, \mathbf{z}^{(u)}\right]^\top$. The latent groups within $\mathbf{z}^{(s)}$ are informed by labels, while the groups in $\mathbf{z}^{(u)}$ are inferred without supervision. We propose to optimize the following lower bound of evidence $p_{\boldsymbol{\xi}}(\mathbf{x}, \mathbf{u})$:

$$
\log p_{\boldsymbol{\xi}}(\mathbf{x}, \mathbf{u}) \geq \underbrace{\mathbb{E}_{q_{\boldsymbol{\psi}}(\mathbf{z}|\mathbf{x})}\left[\log p_{\boldsymbol{\xi}}(\mathbf{x} \mid \mathbf{z})\right]}_{\text{Neural Reconstruction}} + \underbrace{\mathbb{E}_{q_{\boldsymbol{\psi}}(\mathbf{z}|\mathbf{x})}\left[\log p_{\boldsymbol{\xi}}(\mathbf{u} \mid \mathbf{z}^{(s)})\right]}_{\text{Weak Label Supervision}}
$$
$$
- \underbrace{\mathbb{D}_{\mathrm{KL}}\left(q_{\boldsymbol{\psi}}(\mathbf{z} \mid \mathbf{x}, \mathbf{u}) \,\|\, p(\mathbf{z})\right)}_{\text{Prior Regularization}} - \beta \underbrace{\mathbb{D}_{\mathrm{KL}}\left(q_{\boldsymbol{\psi}}(\mathbf{z}) \,\bigg\|\, \prod_{g=1}^{G} q_{\boldsymbol{\psi}}(\mathbf{z}_g)\right)}_{\text{Partial Correlation}}, \tag{2}
$$

where the probabilistic encoder $q_{\boldsymbol{\psi}}(\cdot)$ and decoder $p_{\boldsymbol{\xi}}(\cdot)$ are parameterized by $\boldsymbol{\psi}$ and $\boldsymbol{\xi}$, respectively. $q_{\boldsymbol{\psi}}(\mathbf{z}) = \sum_{n=1}^{N} q_{\boldsymbol{\psi}}\left(\mathbf{z} \mid \mathbf{x}^{(n)}\right) q\left(\mathbf{x}^{(n)}\right)$ is the aggregated posterior over a sample set of size $N$, $g \in \{1, 2, \ldots, G\}$ denotes the latent group index, and hyperparameter $\beta$ controls the penalty scale. We note that the group-wise factorized density in the partial correlation (PC) term is intractable, thus we use the importance sampling (IS) estimator (Li et al., 2023; 2025) to approximate the PC during training. Importantly, the use of weak-supervision labels and the PC penalty here does not degrade the neural reconstruction quality, as verified by the results in Section 4.

## 3.2 MUTUAL INFORMATION MAXIMIZATION GUIDANCE

Given an inferred neural latent $\mathbf{z}$, the next goal is to characterize the visual–semantic factors encoded by a specific latent group $\mathbf{z}_g$. By perturbing the neural latent $\mathbf{z}_g$, we observe the resulting changes in the generated images, which helps reveal the underlying semantic variations. Traditionally, this is done by learning a neural-to-image decoder that directly maps the perturbed latent to an image. However, such a decoder often captures only the most dominant patterns and may miss subtle but meaningful variations in the latent. Instead, we use diffusion to generate an image that maximizes its mutual information with the perturbed latent. This encourages the synthesized image to preserve all relevant information encoded in the latent, rather than averaging it into a single reconstruction. As a result, the generated image better reflects the semantic changes introduced by the perturbation. In this sense, maximizing mutual information provides a more informative guidance signal than directly mapping from $\mathbf{z}$ to $\mathbf{y}$ using a standard decoder.

**Mutual Information-Maximization Guidance.** We start with an original image $\mathbf{y}$ and its corresponding neural latent group $\mathbf{z}_g$. To visualize the semantic features encoded in $\mathbf{z}_g$, we perturb it as $\tilde{\mathbf{z}}_g = \mathbf{z}_g + \gamma \mathbf{1}; \gamma \sim \mathcal{U}(-a, a)$, where $a$ is a predefined half-range and $\mathbf{1} \in \mathbb{R}^{d_g}$ is an all-ones vector matching the dimension of the $g$-th latent group. This perturbation moves $\mathbf{z}_g$ in both positive and negative directions. We then add noise to the image $\mathbf{y}$ and use the diffusion model to gradually denoise it, while guiding the process so that the generated image stays closely related to $\tilde{\mathbf{z}}_g$. Because $\tilde{\mathbf{z}}_g$ is slightly perturbed from the original latent $\mathbf{z}_g$, the new image $\tilde{\mathbf{y}}$ is also a modified version of $\mathbf{y}$. The difference between the two images shows the semantic change caused by moving in the neural latent space. Concretely, we use classifier-guided diffusion (Eq. 1), since its explicit conditional gradient term makes scientific interpretation more accessible. We aim to approximate the two terms on the RHS of Eq. 1. The unconditional score term is estimated using a denoiser $\epsilon_\theta(\mathbf{y}_t, t)$. We then introduce an MI-based guidance to construct the classifier, from which we derive the conditional score $\nabla_{\mathbf{y}_t} \log p(\mathbf{z} \mid \mathbf{y}_t)$. Formally, the MI between $\mathbf{z}_g$ and $\mathbf{y}$ is given by:

$$
\mathbf{MI}(\mathbf{z}_g, \mathbf{y}) = \mathbb{E}_{p(\mathbf{z}_g, \mathbf{y})}\left[\log \frac{p(\mathbf{z}_g, \mathbf{y})}{p(\mathbf{z}_g)p(\mathbf{y})}\right] = \mathbb{E}_{p(\mathbf{z}_g, \mathbf{y})}\left[\log \frac{p(\mathbf{y} \mid \mathbf{z}_g)}{p(\mathbf{y})}\right]. \tag{3}
$$

Note that mutual information captures the full statistical dependence between $\mathbf{z}_g$ and the image $\mathbf{y}$. During diffusion sampling, given the perturbed $\tilde{\mathbf{z}}_g$, maximizing mutual information guides the

process so that the synthesized image $\tilde{\mathbf{y}}$ reflects the semantic information encoded in $\tilde{\mathbf{z}}_g$. Based on Eq. 1, the classifier-guided conditional score becomes:

$$\nabla_{\mathbf{y}_t} \log p^{\eta} \left(\mathbf{y}_t \mid \mathbf{z}_g\right) = \nabla_{\mathbf{y}_t} \log p_{\boldsymbol{\theta}}(\mathbf{y}_t) + \eta \nabla_{\mathbf{y}_t} \mathbf{MI}\left(\mathbf{z}_g, \mathbf{y}_t\right). \tag{4}$$

**Estimation of the Group Mutual Information.** However, the mutual information term in Eq. 4 is intractable and notoriously difficult to compute in practice (Hjelm et al., 2018). According to InfoNCE (Oord et al., 2018), we approximate the density ratio portion $\frac{p(\mathbf{y}|\mathbf{z}_g)}{p(\mathbf{y})}$ in Eq. 3 using a neural network $s_{\boldsymbol{\phi}}\left(\mathbf{z}_g, \mathbf{y}\right)$. To construct the InfoNCE loss, we need to get positive and negative samples for $\mathbf{z}_g$. For an image $\mathbf{y}$, a positive sample $\mathbf{z}_g$ comes from $q_{\boldsymbol{\phi}}(\mathbf{z}_g \mid \mathbf{x})$ where $\mathbf{x}$ is the corresponding neural signal for $\mathbf{y}$, while a negative sample comes from $q_{\boldsymbol{\phi}}(\mathbf{z}_g \mid \hat{\mathbf{x}})$ with $\hat{\mathbf{x}}$ unrelated to $\mathbf{y}$. During training, for each $\mathbf{y}$ and batch size $B$, we collect $\mathcal{Z}_g = \{\mathbf{z}_g^{(1)}, \dots, \mathbf{z}_g^{(B)}\}$, where $\mathbf{z}_g^{(1)}$ is the positive sample and $\mathbf{z}_g^{(i)}$ ($i \in \{2, \dots, B\}$) are negative samples. The noise-contrastive loss $\mathcal{L}_{\mathrm{N}}$ is optimized as follows:

$$\mathcal{L}_{\mathrm{N}}\left(\boldsymbol{\phi}\right) = -\mathbb{E}_{p(\mathbf{z}_g, \mathbf{y})} \left[ \log \frac{\exp\left(s_{\boldsymbol{\phi}}\left(\mathbf{z}_g^{(1)}, \mathbf{y}\right)\right)}{\sum_{\mathbf{z}_g^{(i)} \in \mathcal{Z}_g} \exp\left(s_{\boldsymbol{\phi}}\left(\mathbf{z}_g^{(i)}, \mathbf{y}\right)\right)} \right]. \tag{5}$$

Note that $-\mathcal{L}_{\mathrm{N}}$ provides a lower bound (Oord et al., 2018) on the group-wise mutual information, i.e., $\mathbf{MI}(\mathbf{z}_g, \mathbf{y}) \geq \log(B) - \mathcal{L}_{\mathrm{N}}$. $s_{\boldsymbol{\phi}}\left(\mathbf{z}_g, \mathbf{y}\right)$ specifically approximates the density ratio component of the mutual information $\mathbf{MI}(\mathbf{z}_g, \mathbf{y})$ for the $g$-th latent group. Since the RHS in Eq. 5 provides a normalized probability function, we have:

$$\nabla_{\mathbf{y}_t} \log p_{\boldsymbol{\phi}}\left(\mathbf{z}_g \mid \mathbf{y}_t\right) = -\nabla_{\mathbf{y}_t} \mathcal{L}_{\mathrm{N}}\left(\boldsymbol{\phi}\right) \approx \nabla_{\mathbf{y}_t} \mathbf{MI}\left(\mathbf{z}_g, \mathbf{y}_t\right). \tag{6}$$

During MI-guided image synthesis, at each step $t$, we first estimate its noise-free approximation by $\hat{\mathbf{y}}_0 = \left(\mathbf{y}_t - \sqrt{1 - \alpha_t}\, \boldsymbol{\epsilon}_{\theta}(\mathbf{y}_t, t)\right) / \sqrt{\alpha_t}$, which is then used as input to the $s_{\boldsymbol{\phi}}\left(\cdot\right)$ network. Fig. 2(B) shows the perturbation of the neural latent and the corresponding images generated under MI guidance. MI steers image synthesis within the first latent group $\mathbf{z}_1$, with the guidance strength controlled by the scale parameter $\gamma$.

## 3.3 SEMANTIC IMAGE EDITING WITH DETERMINISTIC DDIM

In diffusion models, early-step noise perturbations primarily distort semantic attributes (e.g., object and category identity) of the clean image, while largely preserving its structural information (e.g., layout, contours, and color composition) (Choi et al., 2022; Wu et al., 2023). As our goal is to uncover the conceptual semantic features encoded within specific neural latent groups, we employ the image editing approach (Meng et al., 2021). This method stops the noise-perturbation process at an intermediate timestep $t = t' \in (0, T)$ and then initiates the backward synthesis process from that step. This approach is employed to retain the basic structure of the reference image. Otherwise, these foundational components will also be generated from pure noise, making it difficult to interpret the effect of neural semantic features built upon them. Furthermore, to ensure that our interpretation is not compromised by the stochastic sampled noise from the standard reverse process, we use the deterministic DDIM sampler.

Formally, we employ a two-stage deterministic image synthesis process. First, we take an original image $\mathbf{y}_0$ and apply deterministic DDIM Inversion (Song et al., 2020; Mokady et al., 2023) using the diffusion denoiser $\boldsymbol{\epsilon}_{\theta}(\cdot)$ from $t = 0$ up to a predefined timestep $t = t'$. This inversion timestep is calibrated to corrupt the semantic attributes of the original image while preserving its structural information. In the second stage, we reverse the process from $t = t'$ to $t = 0$ using a classifier-guided, deterministic DDIM sampling. This reverse procedure is guided by our proposed MI maximization objective, ultimately yielding the synthesized image. This overall process is illustrated in Fig. 2(A). The complete synthesis procedure and a detailed algorithm are provided in Appendix B.

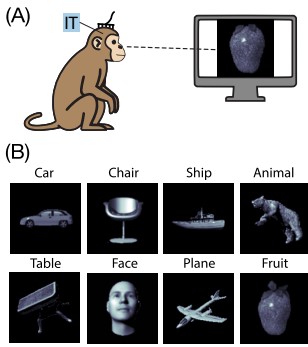

Figure 3: **Macaques IT cortex dataset.** (A) Experimental Setting. (B) Example images from the eight object categories in the dataset.

## 4 EXPERIMENTS

**Macaque IT Cortex Dataset.** We use single-unit spiking responses from the IT cortex of two macaques (M1 and M2) during a passive

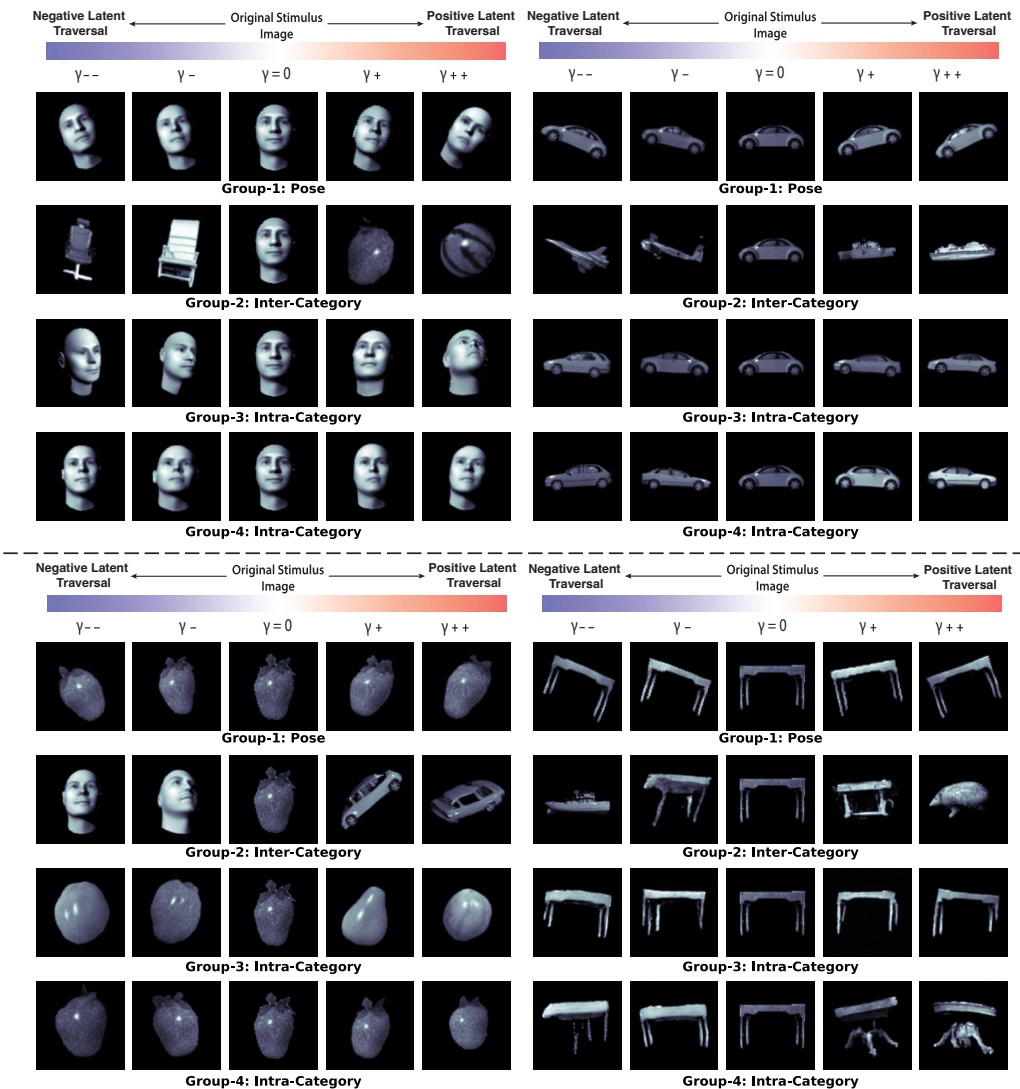

Figure 4: **Synthesized images by MIG-Vis under varying guidance strengths**. Results using a frontal-view face, car, strawberry, and table as the original images.

object recognition task (Majaj et al., 2015). Each head-fixed macaque passively viewed grayscale naturalistic object images while electrophysiological activity was recorded from 58 IT channels in M1 and 110 in M2. Each image was shown for 100 ms at the center of gaze, and neural responses were measured within a 70–170 ms post-stimulus window. The dataset contains 5760 images spanning eight basic-level categories (Fig. 3). To introduce identity-preserving variation, images were generated via ray-tracing with randomized object position, scale, and pose. Following prior work (Lindsey & Issa, 2024), we segment the foreground in each image to reduce background effects.

**Model Settings. (1)** Neural VAE. We adopt a two-layer MLP for both the encoder and the decoder. We set the neural latent dimension number to $D = 24$ and partition it into $G = 4$ latent groups, each with a group rank of $D_g = 6$. The first two groups are supervised: Group 1 is informed by the 3D rotation angles provided in the dataset, and Group 2 uses the 8-way one-hot category_id labels. The remaining groups, Group 3 and Group 4, are learned in an unsupervised manner. **(2)** Neural Encoder. We first extract the DINO embeddings (Caron et al., 2021) from the raw images. For DINO, we adopt the `ViT-B/16` architecture and its image embedding size is 384. The density ratio estimator $s_\phi(\mathbf{z}_g, \mathbf{y})$ is implemented as a three-layer convolutional neural network. **(3)** Diffusion model. We downsample images to a resolution of $128 \times 128$ and trained an image diffusion model based on a U-Net architecture (Ronneberger et al., 2015). The diffusion step $T$ is set as 150 and the $t'$ is set as 135. Further details on the model architecture and hyper-parameters are listed in the Appendix A.

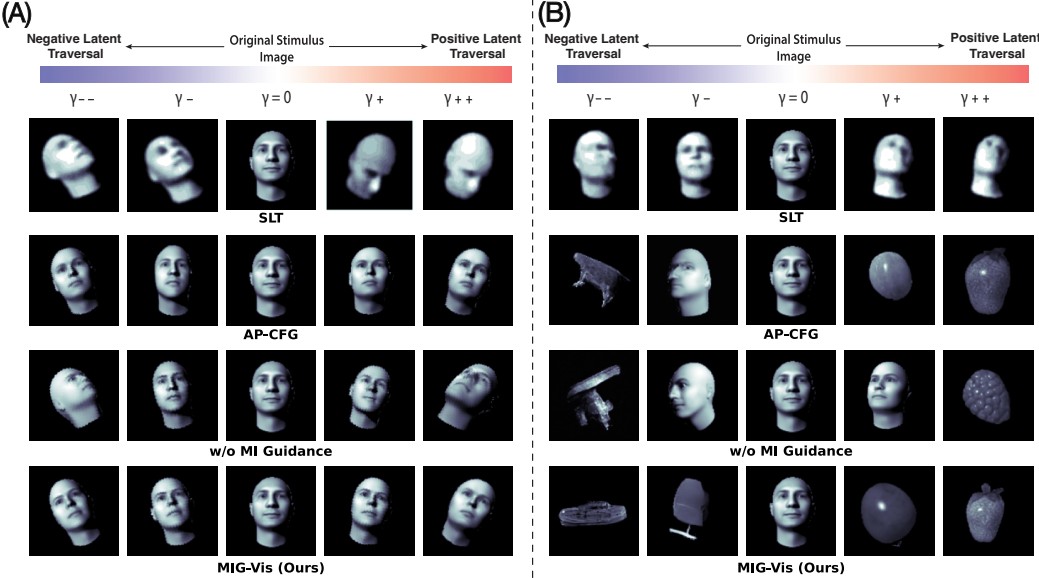

Figure 5: **Comparison of images synthesized by MIG-Vis and baseline methods**. **(A)** Results from probing Latent Group 1 (pose). **(B)** Results from probing Latent Group 2 (inter-category).

## 4.1 DIFFUSION VISUALIZATION OF SEMANTIC SELECTIVITY IN LATENT GROUPS

To perform diffusion-based visualization with MIG-Vis, we select two key components: an **original image** $y_0$, which serves as the starting point for the editing process described in Section 3.3, and a **neural latent group** $z_g$, which we perturb for MI-guided synthesis. We generate images using MIG-Vis under different guidance strengths, $\gamma \in \{-10, -5, 5, 10\}$. In Fig. 4, a frontal-view face, a car, a strawberry, and a table are used as the original images. For each case, we perturb the corresponding neural latent with both positive and negative $\gamma$. The resulting visual-semantic selectivity is summarized as follows:

**(1) Latent Group 1: Pose.** Probing Latent Group 1 primarily modulates pose-related features, such as the rotation of faces and cars. This aligns with the rotation supervision applied to this group's dimensions. Importantly, object category remains unchanged across all synthesized images, indicating that this latent group separates pose variation from other semantic content.

**(2) Latent Group 2: Inter-Category Semantic Attributes.** Notably, despite being supervised only with high-level category identity and no explicit semantic features, Latent Group 2 learns to control inter-category semantic attributes. For example, our MI-guidance transforms the face image into a strawberry. We also observe a direct relationship between activation strength and semantic distance; a larger magnitude of $\gamma$ results in a generated image that is more semantically distinct from the original, demonstrating that the axes of this group subspace encode high-level categorical information.

**(3) Latent Group 3 and 4: Intra-Category Content Details.** Latent Groups 3 and 4, discovered without supervision, both encode intra-category content variations. Group 3 primarily modulates the appearance of faces and strawberries with little effect on cars, whereas Group 4 significantly alters cars and tables while minimally affecting faces. This selective behavior likely reflects distinct patterns of visual variation across object categories. These results suggest that the neural latent manifold in these dimensions is locally structured rather than globally aligned. Different object categories occupy different regions of the manifold and vary along different directions. Geometrically, this implies that the manifold is anisotropic and curved: the principal directions of variation are category-dependent. Faces and strawberries extend strongly along the dimensions of Group 3, whereas cars and tables extend more along Group 4. Thus, intra-category variation is organized into separate local tangent directions rather than a single shared global axis across all objects.

## 4.2 Neural Guidance Baseline Methods for Comparison

As prior works are designed for different types of neural data (e.g., fMRI), here we isolate their core neural guidance approaches during image synthesis for a fair comparison with our method:

• **Standard Latent Traversal (SLT)**: a standard approach for latent variable manipulation (Chen et al., 2018; Esmaeili et al., 2019), which trains a decoder to map neural latents to images. We perform latent traversal with $\tilde{\mathbf{z}}_g = \mathbf{z}_g + \gamma \mathbf{1}$.

• **Activation Probing via Classifier-Free Guidance (AP-CFG)**: the neural manipulation method used in **BrainACTIV** (Cerdas et al., 2024), which synthesizes images guided to maximize or minimize the activation of a target cortical region. It employs a classifier-free guidance diffusion (Ho & Salimans, 2022) that jointly models the denoising and guidance gradients.

• **Ours w/o MI Guidance**: We perform latent traversal by shifting the group latent as $\tilde{\mathbf{z}}_g = \mathbf{z}_g + \gamma \mathbf{1}$. Instead of mutual information, we train a probabilistic encoder that maps an image $\mathbf{y}$ to the neural latent $\mathbf{z}_g$, i.e., learning $p(\mathbf{z}_g|\mathbf{y})$. Given $\tilde{\mathbf{z}}_g$, we generate a new image $\tilde{\mathbf{y}}$ via diffusion, guided by $\nabla_{\mathbf{y}} \log p(\mathbf{z}_g|\mathbf{y})$, which encourages the generated image to produce a latent with highest likelihood matching $\tilde{\mathbf{z}}_g$.

**Comparison Results Analyses.** In Fig. 5, we compare MIG-Vis with the three baselines introduced above, using a frontal-view face as the original image. We focus on two representative latent groups for this analysis: Group 1 and Group 2, which have been identified as controlling pose and inter-category factors, respectively. We first observe that images manipulated by **SLT** exhibit some rotation when probing Group 1, but the changes are not as clean as those produced by other methods. When probing Group 2, the object category remains unchanged. This suggests a limitation of decoder-based image generation compared to diffusion-based generation. **AP-CFG** performs reasonably well in capturing rotation semantics; however, its perturbations for inter-category variations are less clean compared to our **MIG-Vis**.

Importantly, in the ablation of our framework **w/o MI guidance**, where diffusion is guided by likelihood-based alignment $\nabla_{\mathbf{y}} \log p(\mathbf{z}_g|\mathbf{y})$, we observe that pose-related rotation semantics can be captured moderately well. Both ours with and without MI rely on linear activation probing to define the latent subspace. However, the key difference lies in the guidance objective. Likelihood-based guidance only enforces that the synthesized image be mapped back to the target latent $\mathbf{z}_g$ by the learned probabilistic encoder. In other words, diffusion merely needs to generate an image that the encoder recognizes as having latent $\mathbf{z}_g$. This constraint is one-sided and encoder-dependent: as long as the encoder assigns high likelihood to $\mathbf{z}_g$, the objective is satisfied. For simple, low-dimensional variations such as rotation, this may be sufficient. However, for inter-category semantic features, whose latent structure is complex and nonlinear, this objective becomes too weak. Multiple visually different images may map to similar latent values under the encoder, and subtle semantic structure may be averaged out. As a result, likelihood-based guidance can produce inconsistent or unrealistic categorical transitions.

In contrast, our **MIG-Vis** employs MI guidance, which imposes a stronger constraint. Rather than asking whether the encoder can recognize the image as having latent $\mathbf{z}_g$, MI maximization requires that the synthesized image and $\mathbf{z}_g$ share maximal statistical dependence. Intuitively, likelihood guidance asks: *Can the encoder recognize this image as having latent $\mathbf{z}_g$?* MI guidance asks: *Does this image truly express the information contained in $\mathbf{z}_g$?* The latter is a stricter requirement. It forces the diffusion process to generate images that visibly and structurally reflect the semantic content encoded in the perturbed latent group, rather than merely satisfying an encoder consistency check. Consequently, MI guidance better preserves complex inter-category structure and produces smooth, realistic transitions across object instances.

## 4.3 Fine-Grained Neural Selectivity within Latent Group

In Figs. 6 and 7(A), we probe pair-wise latent dimensions within the Group 3 intra-category subspace by starting from both a face image and a strawberry image, and within the Group 2 inter-category subspace by starting from a table image. We observe that perturbing the same dimension pairs results in different semantic changes across object categories.

In Fig. 6, for dimension pair $[1, 2]$, perturbations applied to the face primarily modulate orientation and gaze direction (e.g., upward gaze or left orientation shift). In contrast, perturbing the same dimensions for the strawberry mainly produces texture smoothing and global lighting changes. Similarly, within the $[3, 5]$ subspace, faces exhibit structural variations such as right-side gaze shifts,

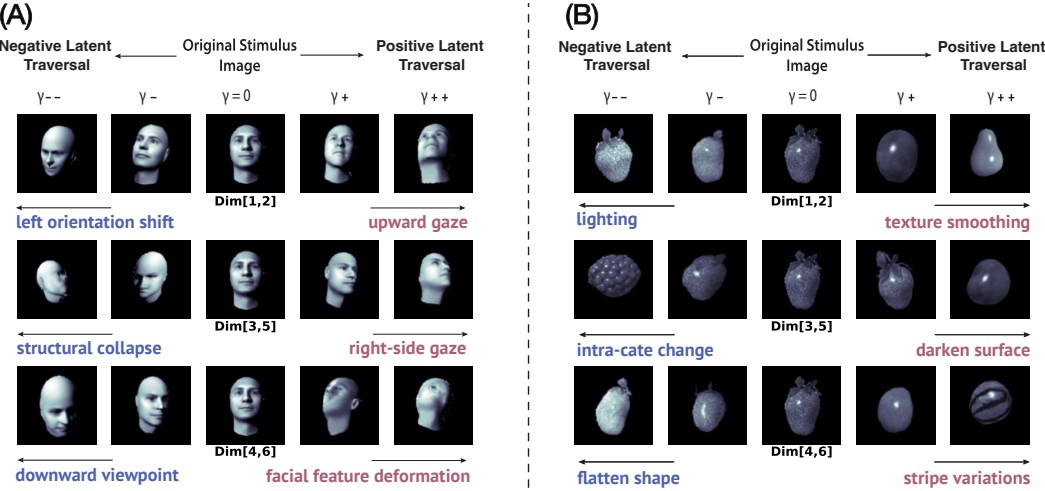

Figure 6: **Pair-wise visual-semantic probing in latent Group 3.** **(A)** MIG-Vis synthesis from a frontal-view face. **(B)** MIG-Vis synthesis from a strawberry. These results show that different dimension pairs within the same latent group encode distinct, fine-grained visual-semantic attributes.

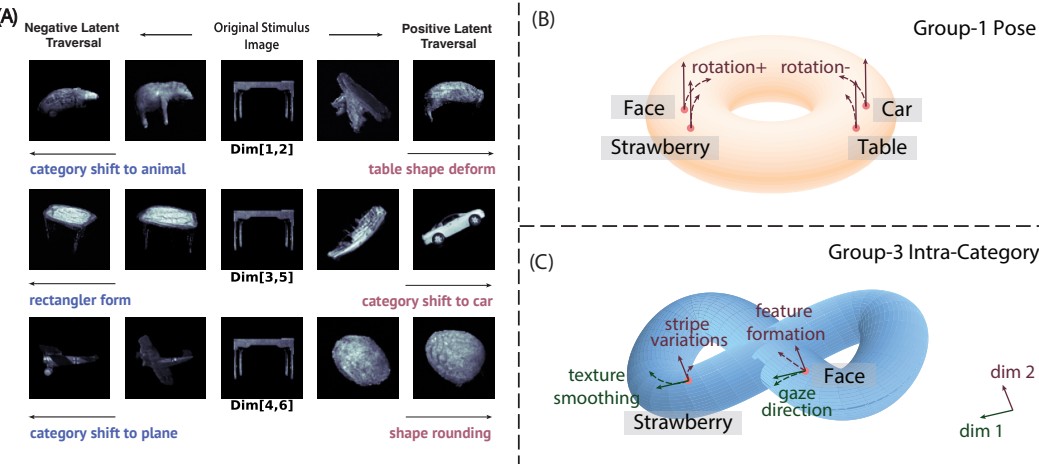

Figure 7: **(A) Pair-wise visual-semantic feature investigation within latent Group 2.** Results of MIG-Vis from probing a table object. **(B) and (C): Neural manifolds of two latent groups.**

whereas strawberries show intra-category changes and surface darkening. For $[4, 6]$, perturbations deform facial features and alter viewpoint in faces, but induce shape flattening and stripe variations in strawberries.

With these semantic probing visualizations produced by MIG-Vis, we gain insight into the structure of the high-dimensional neural space in the higher visual cortex. Here, we illustrate plausible neural manifolds within example subspaces.

For the latent dimensions associated with **Group-1 (pose)**, increasing $\gamma$ produces systematic rotational changes across all object categories (Fig. 4). However, the *direction* of rotation depends on the object: faces rotate clockwise while cars rotate counterclockwise. A similar pattern is observed for strawberries and tables. This indicates that perturbations along these latent dimensions consistently encode a rotation semantic, regardless of object identity. In other words, manipulating the same latent axis always induces rotation, even though the visual manifestation of that rotation may differ across categories. This observation suggests that the underlying geometry of the latent space in these dimensions is structured rather than linear. A natural hypothesis is that the manifold along these dimensions resembles a torus (Fig. 7B). In this view, different object categories occupy different locations on the toroidal manifold: cars and tables lie closer on one side, while faces and strawberries lie closer on the opposite side. When a latent perturbation is applied (vertical arrows

in Fig. 7B), its projection onto the toroidal manifold induces motion along the surface in opposite directions for different object regions. This results in clockwise rotation for some objects and counterclockwise rotation for others. Crucially, despite these differences in local behavior, the semantic meaning of movement along this latent axis remains consistent across categories: it corresponds to object rotation. We therefore interpret these dimensions as forming a **globally consistent semantic manifold** in neural latent space.

In contrast, the latent dimensions associated with **Group-3 (intra-category)** exhibit a fundamentally different behavior. For example, when perturbing a face along dimension 1 versus dimension 2 with changing $\gamma$, we observe distinct semantic changes such as gaze direction and feature formation. However, applying the same perturbations to a different object, such as a strawberry, produces entirely different semantic variations, such as changes in texture smoothing or stripe patterns. This stands in sharp contrast to the Group-1 pose dimensions, where latent perturbations consistently correspond to the same semantic transformation across objects. Here, the same latent directions do not induce a shared semantic effect when applied to different object categories. This suggests that the underlying manifold in this subspace is substantially more complex and nonlinear (e.g., Fig. 7C). Rather than forming a globally structured surface like the toroidal rotation manifold in Group-1, the geometry here appears warped and irregular. Perturbations along different latent axes project onto the manifold in different ways: some may induce motion along a locally structured trajectory (e.g., along an "8-shaped" curve), while others act in directions that are effectively orthogonal to it. As a result, the semantic interpretation of these dimensions becomes object-dependent. Unlike the globally consistent rotation semantics observed in Group-1, no shared transformation emerges across categories. Instead, semantic meaning can only be interpreted locally, relative to each object's position on the manifold. Taken together, this indicates that while some latent subspaces encode globally consistent semantics (e.g., pose), others correspond to highly warped intra-category manifolds whose effects are inherently local rather than universal.

Thus, MIG-Vis serves as an intuitive tool for visualizing neural manifolds and generating hypotheses, while also pointing toward future neuroscience research on formally characterizing the geometry of neural subspaces in higher visual cortex.

## 4.4 Neural Reconstruction Evaluation

To verify that our neural VAE module maintains high-quality neural reconstruction with the partial correlation regularization and weak label supervision, we quantitatively evaluated its performance. Table 1 presents the R-squared ($R^2$ in %) values for both macaques, comparing our module to standard unsupervised VAE. These results demonstrate that the neural reconstruction quality is well-preserved, incurring only a marginal performance drop compared to the standard VAE. We hypothesize that the weakly-supervised labels induce a rotation of the neural latent subspace while preserving the information necessary for reconstruction. From the $R^2$ results of ablation studies on our VAE module's two introduced terms, we observe that each has only a minimal negative impact on neural reconstruction. We also plot the raw neural signals and their reconstruction across dimensions for four selected trials in Fig. 8 in the Appendix A.4.

Table 1: Neural reconstruction quality comparisons on IT cortex dataset. We report the mean and standard deviation of explained variance ($R^2$ in %) over five runs.

| Subject | Method | $R^2(\%) \uparrow$ |
|---|---|---|
| M1 | Standard VAE | 78.62 ($\pm$0.58) |
| | Ours w/o Sup. | 76.90 ($\pm$0.53) |
| | Ours w/o PC. | 77.30 ($\pm$0.62) |
| | **Ours** | 76.58 ($\pm$0.64) |
| M2 | Standard VAE | 83.72 ($\pm$0.47) |
| | Ours w/o Sup. | 82.39 ($\pm$0.59) |
| | Ours w/o PC. | 82.21 ($\pm$0.55) |
| | **Ours** | 81.86 ($\pm$0.51) |

## 5 Conclusion

Our contributions can be summarized into the following points: (1) Leveraging advanced neural latent variable models, this is the first work to explore neural representations with semantic selectivity in the higher visual cortex from electrophysiological data. (2) To interpret the visual-semantic features encoded within a specific neural latent group, we use a DDIM-based deterministic image editing approach with a proposed mutual information maximization objective. (3) The edited stimuli faithfully demonstrate distinct and high-level visual features, verifying the semantic selectivity within these inferred neural latent groups. Our work offers a critical step toward understanding the compositional, multi-dimensional nature of visual coding in the primate visual cortex.

ACKNOWLEDGEMENT

This work was supported by the seed grant from Microsoft GenAI.

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

APPENDIX

# A  DETAILS OF THE MIG-VIS METHOD

## A.1  MODEL TRAINING DETAILS

For the partially disentangled neural VAE, we employ an architecture consisting of a two-layer MLP for both the probabilistic encoder and the probabilistic decoder networks. Each MLP uses ReLU activations. For the neural latent dimension number $k$ to both the two macaques, we set $k = 24$ and partition it into four distinct latent groups, each of dimension 6 (i.e., group rank equals to 6). This group rank setting provides the expressiveness to model complex visual semantic factors (e.g., inter-category variations) within a unique latent group. For latent group 1, we weakly supervise its learning with the 3D rotation angles of the object provided in the dataset; for latent group 2, we use the 8-way category identity labels (e.g., "Animals" as integer label 0, "Boats" as integer label 1); latent groups 3 and 4 are learned in a fully unsupervised manner. We use a learning rate of $1 \times 10^{-3}$ and set partial correlation penalty `pc_weight` to $5 \times 10^{-5}$ on both two macaques. We use the Adam Optimizer Kingma & Ba (2014) for optimization.

For the image diffusion model, we downsample the grayscale images to a resolution of $1 \times 128 \times 128$ and train the diffusion model on the resulting inputs. We augment the dataset using random horizontal flips to improve model generalization. We adopt the architecture of the image diffusion denoiser as the U-Net (Ronneberger et al., 2015). We adopt the $\epsilon$-parameterization in our diffusion model for improved training stability, given the relatively small dataset size and diffusion timestep settings. The diffusion timestep is set as 150. The embedding input dimension to the U-Net architecture is set as 64 and the U-Net has three down-sampling and up-sampling layers. The training batch size is set as 64. We train the U-Net denoiser model on $40,000$ iterations with a learning rate of $2 \times 10^{-4}$. All experiments are conducted using PyTorch on a compute cluster equipped with NVIDIA A40 GPUs. The training phase of the image diffusion model takes approximately 20 hours. We also apply exponential moving average (EMA) with a decay rate of $0.98$ to stabilize training and improve generalization.

## A.2  DERIVATION OF THE NOISE-FREE PREDICTION $\hat{\mathbf{y}}_0$

We here elaborate the approximation of the clean image sample $\hat{\mathbf{y}}_0$, as used in Section 3.2. It is computed from the perturbed data point $\mathbf{y}_t$ and the prediction of the denoiser model $\epsilon_\theta(\mathbf{y}_t, t)$.

As in the forward process of diffusion models, given a clean stimulus image $\mathbf{y}_0$, it generates a noisy sample $\mathbf{y}_t$ by gradually adding Gaussian noise through the following closed-form equation:

$$\mathbf{y}_t = \sqrt{\alpha_t}\, \mathbf{y}_0 + \sqrt{1 - \alpha_t}\, \epsilon, \quad \epsilon \sim \mathcal{N}(\mathbf{0}, \mathbf{I}). \tag{7}$$

The above re-parameterized forward process represents $\mathbf{y}_t$ as a linear combination of the clean image $\mathbf{y}_0$ and the Gaussian noise $\epsilon$. By rearranging the above equation, we obtain an exact expression for the original image sample $\mathbf{y}_0$:

$$\mathbf{y}_0 = \frac{1}{\sqrt{\alpha_t}} \left( \mathbf{y}_t - \sqrt{1 - \alpha_t}\, \epsilon \right). \tag{8}$$

As the true noise $\epsilon$ is unknown during the sampling phase, it is approximated by the learned denoiser neural network $\epsilon_\theta(\mathbf{y}_t, t)$. Inserting this estimate into the expression above provides an approximation of the clean image $\mathbf{y}_0$:

$$\hat{\mathbf{y}}_0 = \frac{\mathbf{y}_t - \left( \sqrt{1 - \alpha_t} \right) \epsilon_\theta(\mathbf{y}_t, t)}{\sqrt{\alpha_t}},$$

this formula is used consistently across both DDPM and DDIM frameworks to obtain a noise-free approximation of the original image from the noisy sample at each timestep.

## A.3  GROUP-WISE DISENTANGLED NEURAL LATENT SUBSPACE INVESTIGATION

To ensure that the neural VAE module within the MIG-Vis framework maintains high-quality neural reconstruction along with the partial correlation disentanglement regularization and weakly supervised label guidance, we present the quantitative neural reconstruction results of both two macaques

in Table 2, which records the R-squared values ($R^2$, in %) and RMSE of our module and the standard unsupervised VAE. These results indicate that the neural reconstruction quality is well-preserved, with only a slight performance drop compared to the standard VAE.

A possible explanation is that the weakly-supervised labels rotate the neural latent subspace in a way that preserves most of the information necessary for reconstructing the neural activity. This observation is reasonable, given that these labels can be accurately decoded from the neural activity itself. Furthermore, we evaluate the disentanglement quality of the inferred latent subspace using the widely adopted Mutual Information Gap (MIG) metric (Chen et al., 2018), also reported in Table 2. The results demonstrate that the neural latents learned by MIG-Vis are significantly more disentangled than those of the standard VAE.

Table 2: Performance report of our neural VAE module and a standard VAE on IT cortex neural activity from macaque M1 and M2. We report explained variance ($R^2$), root mean squared error (RMSE), and mutual information gap (MIG) to assess reconstruction accuracy and latent disentanglement. Results are averaged over 5 runs; bold numbers indicate the highest MIG score on each subject.

| Metrics \ Method | M1 | | M2 | |
|---|---|---|---|---|
| | Standard VAE | **Ours** | Standard VAE | **Ours** |
| $R^2(\%) \uparrow$ | 78.62 ($\pm$0.58) | 76.58 ($\pm$0.64) | 83.72 ($\pm$0.47) | 81.86 ($\pm$0.51) |
| RMSE $\downarrow$ | 48.49 ($\pm$0.39) | 50.47 ($\pm$0.39) | 40.77 ($\pm$0.28) | 43.44 ($\pm$0.35) |
| MIG($\%$) $\uparrow$ | 33.27 ($\pm$0.82) | **44.23** ($\pm$0.61) | 28.65 ($\pm$0.71) | **49.85** ($\pm$0.55) |

Table 3: Explained variance of neural signals from the supervised and unsupervised neural latents of the two macaques M1 and M2.

| Subject | Metrics | Full Latents | Supervised Latents | Unsupervised Latents |
|---|---|---|---|---|
| M1 | $R^2$ (%) $\uparrow$ | 76.13 ($\pm$0.27) | 32.50 ($\pm$0.19) | 20.88 ($\pm$0.25) |
| | RMSE $\downarrow$ | 48.86 ($\pm$0.21) | 80.75 ($\pm$0.28) | 88.95 ($\pm$0.29) |
| M2 | $R^2$ (%) $\uparrow$ | 71.82 ($\pm$0.28) | 28.11 ($\pm$0.22) | 19.63 ($\pm$0.17) |
| | RMSE $\downarrow$ | 53.09 ($\pm$0.16) | 84.79 ($\pm$0.32) | 89.65 ($\pm$0.31) |

A.4 NEURAL SIGNAL RECONSTRUCTION

## B DETAILED MUTUAL-INFORMATION MAXIMIZATION GUIDANCE ALGORITHM

---

**Algorithm 1** Synthesize Stimulus Image for Semantic Latent Group Discovery

---

**Input:** Target Neural Latent Group $\mathbf{z}_g$, Guidance Scale $\eta$, Sampling Step $t'$
**Output:** Synthesized Image Stimulus with Guidance $\tilde{\mathbf{y}}_0$

---

1 Initiate Original Image Stimulus $\mathbf{y}_0$
2 **for** $t = 1$ **to** $t'$ **do**
3 $\quad$ $\mathbf{y}_t = \sqrt{\frac{\alpha_t}{\alpha_{t-1}}} \cdot \mathbf{y}_{t-1} + \left(\sqrt{\frac{1}{\alpha_t} - 1} - \sqrt{\frac{1}{\alpha_{t-1}} - 1}\right) \cdot \boldsymbol{\epsilon}_\theta(\mathbf{y}_{t-1}, t)$ $\qquad$ $\triangleright$ DDIM Inversion
4 **for** $t = t'$ **to** $1$ **do**
5 $\quad$ $\hat{\boldsymbol{\epsilon}}(\mathbf{y}_t, t) = \boldsymbol{\epsilon}_\theta(\mathbf{y}_t, t) + \eta \nabla_{\mathbf{y}_t} \mathcal{L}_N(\boldsymbol{\phi})$ $\qquad$ $\triangleright$ Mutual Information Guided Gradient
$\quad$ $\mathbf{y}_{t-1} = \sqrt{\frac{\alpha_{t-1}}{\alpha_t}} \cdot \mathbf{y}_t + \left(\sqrt{\frac{1}{\alpha_{t-1}} - 1} - \sqrt{\frac{1}{\alpha_t} - 1}\right) \cdot \hat{\boldsymbol{\epsilon}}(\mathbf{y}_t, t)$ $\qquad$ $\triangleright$ Guided DDIM Sampling

---

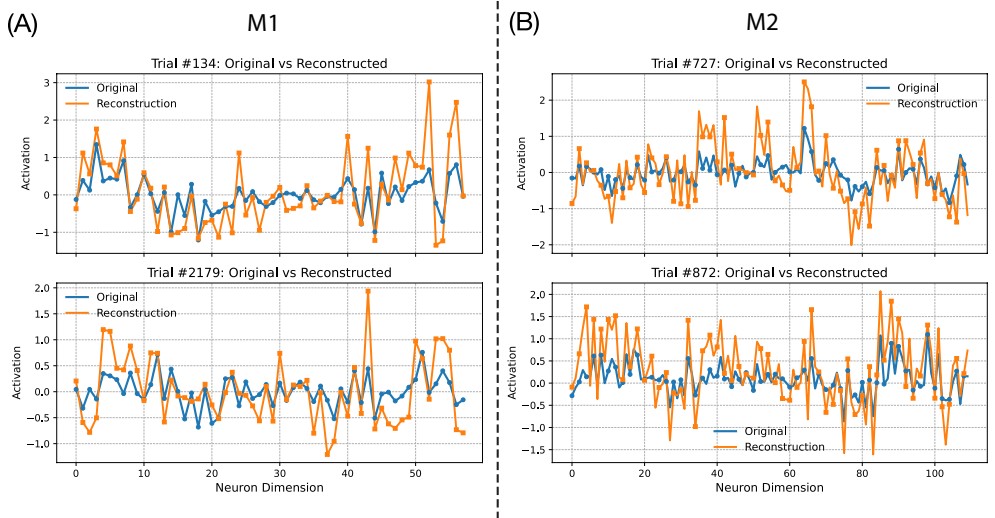

Figure 8: **(A)** Raw Neural signals and the reconstructions from the neural LVM module in MIG-Vis on the M1 subject. **(B)** Raw Neural signals and the reconstructions from the neural LVM module in MIG-Vis on the M2 subject.

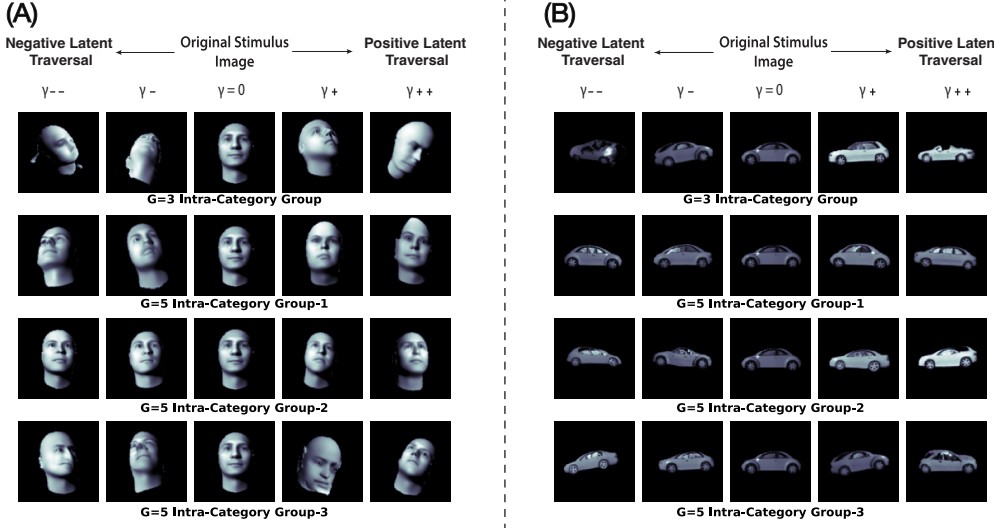

Figure 9: **Neural latent group number** $G$ **analyses.** **(A)** Synthesized images of MIG-Vis from probing a frontal-view human face. **(B)** Synthesized images of MIG-Vis from probing a car object.

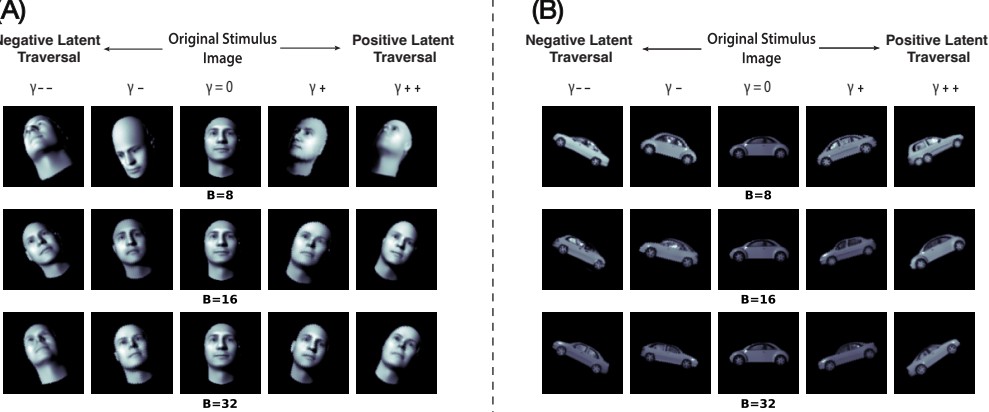

Figure 10: **InfoNCE negative sample** $B$ **analyses in Group-1 (Pose).** **(A)** Synthesized images of MIG-Vis from probing a frontal-view human face. **(B)** Synthesized images of MIG-Vis from probing a car object.

