# OpenReview forum: "Uncovering Semantic Selectivity of Latent Groups in Higher Visual Cortex with Mutual Information-Guided Diffusion"
_ICLR.cc/2026/Conference — ICLR 2026 Poster_

### Official Review · Reviewer_4PLU · 2025-10-29

**Soundness:** 3
**Presentation:** 3
**Contribution:** 2
**Rating:** 6
**Confidence:** 4

**Summary:**

This paper presents MIG-Vis, a computational method for interpreting mixed selectivity in neural population dynamics. The approach combines a semi-supervised variational autoencoder that decomposes neural latent representations with a diffusion model for stimulus generation. The method replaces variance-based optimization with mutual information maximization using InfoNCE estimation to generate visual stimuli that selectively activate specific latent factors. Experimental validation is conducted on neural recordings from macaque IT cortex during visual decision-making tasks, with reported improvements in disentanglement metrics and demonstration of generated stimuli showing selectivity to targeted neural factors.

**Strengths:**

1. The proposed mutual information maximization approach offers a new perspective for stimulus generation in neural interpretation tasks, demonstrating improved disentangled visual stimulus reconstruction effects.
2. The paper is well-structured with relevant background coverage, providing sufficient implementation detail for reproducibility.

**Weaknesses:**

1. The paper reports poor reconstruction R² values alongside high MIG disentanglement scores, creating a concerning contradiction. If the VAE cannot adequately explain neural variance, the validity of downstream stimulus generation becomes questionable.
2. The entire evaluation relies solely on a single real-world dataset (macaque IT cortex) with no additional validations such as synthetic data. This provides insufficient evidence for method generalizability or robustness.
3. The lack of controlled synthetic datasets with known underlying factors prevents proper validation of whether the method recovers true latent structure versus dataset-specific artifacts.
4. The approach bears high similarity to the authors' prior BeNeDiff method, which represents a natural and important baseline for comparison. Additional comparison would help substantiate the claimed effectiveness of MI guidance over variance-based approaches.

**Questions:**

1. Given the suboptimal reconstruction performance of VAE, how can you demonstrate that the learned representations capture meaningful neural dynamics rather than statistical artifacts?
2. Can you provide ablation studies isolating the contribution of MI guidance versus VAE architecture choices to determine which components actually drive the reported improvements?

Suggestions:
1. The authors should include at least one synthetic dataset with known ground truth factors to provide reliable evidence for method effectiveness and generalization capabilities.
 2. The authors should provide explicit quantitative comparison with their prior BeNeDiff method to clearly demonstrate the specific advantages of MI guidance over variance-based approaches.

---

> ### Author Response · Authors · 2025-11-22
>
> Dear Reviewer 4PLU,
>
> Thank you for your insightful comments and the recognition of our work. We provide clarifications and new experimental results to address your questions below.
>
> > Given the suboptimal R² reconstruction performance of VAE, how can you demonstrate that the learned representations capture meaningful neural dynamics rather than statistical artifacts?
>
> We appreciate the question regarding the raw neural signal reconstruction performance ($R^2$) for the neural LVM module in our proposed MIG-Vis framework. We show the quantitative metrics in Table 1 of the updated manuscript. We would like to clarify that the reconstruction quality of our Neural LVM is robustly maintained and is comparable to results achieved by the standard VAE. Specifically, on both two primates, our neural LVM module explains more than 75% of the overall neural signal variance. Furthermore, we have added experiments illustrating the raw neural signals and their reconstruction across dimensions for four selected trials in Figure 8 of the revised manuscript. We observe that our Neural VAE module reconstructs the neural signals with high-fidelity, which verifies that our inferred neural latents faithfully capture the underlying neural encodings.
>
> > The approach bears high similarity to the BeNeDiff method, which represents a natural and important baseline for comparison. Additional comparison would help substantiate the claimed effectiveness of MI guidance over variance-based approaches.
>
> This is a great suggestion regarding baseline comparison. We agree that BeNeDiff [1] is an important related work that uses generative diffusion models to explore neural dynamics. However, a direct comparison of BeNeDiff to our proposed MIG-Vis framework here is not feasible due to the differences in neural data types.
>
> We note that the BeNeDiff method is designed for temporal neural trajectories, in which its guidance objective is to maximize the trajectory variance over timesteps to synthesize video results. In contrast, the model framework as well as the IT cortex dataset [2] we use consists of neural signals where each trial corresponds to a single time point. Since there is no temporal dimension or dynamics in our data, the variance maximization objective in BeNeDiff cannot be directly adopted in this setting.
>
> On the other hand, given that the variance objective in BeNeDiff corresponds to a specific order of moment of the neural data, we can compare MIG-Vis with a baseline using only the first-order (mean-value) moment as the maximization objective. The results are presented in row 4 of Figure 5 in the revised manuscript. Their comparison of synthesized images in row 5 of Figure 5 demonstrates that our proposed mutual information guidance (MIG) objective leads to high-fidelity and visual-semantically interpretable visualizations (row-5 in Figure 5).
>
> > Can you provide ablation studies isolating the contribution of MI guidance versus VAE architecture choices to determine which components actually drive the reported improvements?
>
> This is an important concern for validating the actual contribution of our proposed Mutual Information Guidance (MIG) objective. We conducted ablation studies where we replaced the MIG objective with a simpler mean-value activation guidance objective. The results are presented in row 4 of Figure 5 in the revised manuscript. Their comparison of synthesized images in row 5 of Figure 5 demonstrates that our proposed mutual information guidance (MIG) objective leads to high-fidelity and visual-semantically interpretable visualizations (row-5 in Figure 5).
>
> > The authors should include at least one synthetic dataset with known ground truth factors to provide reliable evidence for method effectiveness and generalization capabilities.
>
> This is a key concern regarding the generalizability of our proposed MIG-Vis framework across datasets.
>
> We acknowledge that the generalizability of the proposed MIG-Vis across different datasets is a key concern. We note that the synthesis of an orchestrated dataset that includes both the stimulus image and the neural signals is challenging to truly reflect the real-world joint distribution of these two modalities of data. We also note that prior works focusing on scientific discovery with diffusion synthesized images are typically evaluated on a single, high-quality dataset [3].
>
>
>
> We welcome any additional questions and look forward to your feedback.
>
> Refs:
>
> [1] Exploring Behavior-Relevant and Disentangled Neural Dynamics with Generative Diffusion Models. (Wang et al., 2024)
>
> [2] Simple learned weighted sums of inferior temporal neuronal firing rates accurately predict human core object recognition performance. (Majaj et al., 2015)
>
> [3] Cortical discovery using large scale generative models. (Luo et al., 2023)

---

### Official Review · Reviewer_bCGZ · 2025-11-01

**Soundness:** 2
**Presentation:** 2
**Contribution:** 2
**Rating:** 2
**Confidence:** 4

**Summary:**

The paper tackles the enduring question of how information is represented in higher-level visual cortex, a challenge compounded by the mixed selectivity of neurons to multiple latent variables. The authors propose MIG-Vis, which combines a group-wise disentangled variational autoencoder (VAE) with a mutual-information-guided diffusion model to visualize and interpret the semantic content encoded by neural latent groups. Using macaque IT recordings and controlled object stimuli, the method aims to reveal how different latent groups in neural activation space correspond to features such as pose, category, or intra-category content.

**Strengths:**

The paper addresses an important problem—understanding how visual cortex encodes semantically meaningful subspaces—using a technically interesting and timely approach. The combination of a group-wise disentangled VAE and mutual information-guided diffusion is conceptually elegant.

**Weaknesses:**

- While the idea is strong, the paper remains largely qualitative, and it is unclear what new insights about neural coding are gained beyond demonstrating that the model can recover known, supervised factors. Two of the latent groups are trained with supervised labels, presumably corresponding to pose and category. For these, the generated images vary sensibly along the provided latent axes, which serves as a good sanity check for the method. However, it is not at all clear how much variance these axes explain in the overall neural space. Given that a considerable fraction of IT variance is already known to relate to category and pose, these results may simply reflect that expected structure. It would be useful to quantify how much of the explained variance these supervised groups account for, and whether the learned representations capture any additional structure beyond the labeled variables.


- The most interesting aspect of the approach lies in the unsupervised latent groups, yet these are only very briefly discussed and remain largely qualitative. It is unclear what these groups encode, how stable they are across datasets or neurons, and what fraction of neural variance they capture. A more detailed analysis of these unsupervised dimensions would greatly strengthen the paper and help assess whether the model uncovers genuinely new aspects of neural representation.


- Methodologically, the two-step diffusion process—first inverting and then guiding image synthesis—should be motivated more clearly. Is this procedure intended to constrain the generation to remain on the natural image manifold, for instance analogous to preserving phase information in Fourier space? Does the choice of starting image determine the outcome in the second stage?


- Finally, the introduction would benefit from clearer framing, as it currently shifts rapidly between topics without defining a specific gap in knowledge or central research question.

**Questions:**

- Can the authors clarify what the supervised latent groups represent—are these indeed pose and category—and how the corresponding labels were generated? How much of the total neural variance do these supervised groups explain, and do they capture any structure beyond what would be expected from the labeled variables? For the unsupervised groups, what features do they encode, how consistent are they across neurons or datasets, and what proportion of variance do they account for?

-  In terms of methodology, what is the motivation for the two-step diffusion process (inversion followed by guided synthesis)? Is this intended to constrain image generation to remain on the natural image manifold, and does the choice of starting image influence the final outcome?

---

> ### Author Response · Authors · 2025-11-22
>
> Dear Reviewer bCGZ,
>
> Thank you for your insightful comments. We have addressed your points with the following clarifications and new results. We hope these will resolve most of your concerns, and they can be taken into account when evaluating the final review score.
>
> > Can the authors clarify what the supervised latent groups represent—are these indeed pose and category—and how the corresponding labels were generated?
>
> We confirm this understanding. The first two supervised latent groups represent the neural latent variables that account for the pose and category semantic variations. We extract the pose labels and the category identity labels of the stimulus image from the public dataset and conduct preprocess on them [1].
>
> > How much of the total neural variance do these supervised groups explain?
>
> This is a critical point, we have the explained neural variance of the supervised neural latents listed in Table 1 below for your reference. We first note that acoording to the formula of calculating the explained variance $R^2=1-\frac{\sum_i\left(y_i-\hat{y}_i\right)^2}{\sum_i\left(y_i-\bar{y}\right)^2}$, the overall $R^2$(%) is not a simple sum of the supervised latents explained variance and the unsupervised latents explained variance. Nevertheless, as shown in the table below, the supervised latent groups alone on both two primates contribute to a significant portion of the total neural data variance.
>
> | Subject |       Metrics       |    Full Latents    | Supervised Latents |
> | :-----: | :-----------------: | :----------------: | :----------------: |
> |   M1    | $R^2$(%) $\uparrow$ | $76.13 (\pm 0.27)$ | $32.50 (\pm 0.19)$ |
> |   M1    |  RMSE $\downarrow$  | $48.86 (\pm 0.21)$ | $80.75 (\pm 0.28)$ |
> |   M2    | $R^2$(%) $\uparrow$ | $71.82 (\pm 0.28)$ | $28.11 (\pm 0.22)$ |
> |   M2    |  RMSE $\downarrow$  | $53.09 (\pm 0.16)$ |   $84.79(±0.32)$   |
>
> *Table 1: Explained neural variance from the supervised neural latents of two macaques.*
>
> > For the unsupervised groups, what features do they encode, and what proportion of variance do they account for?
>
> We have listed the explained neural variance of the unsupervised neural latents in the following Table 2 for your reference. The results show that the unsupervised latent groups on both two primates also contribute to a substantial portion of the neural data variance, which demonstrate their necessity in neural reconstruction. Furthermore, as shown in Figure 4 of the manuscript (row-3 and row-4 of both (A) and (B)), we uncover that these unsupervised latent groups encode **intra-category content semantic variations**. Specifically, the first unsupervised group primarily modulates the details of the face and chair categories, whereas the second unsupervised group specializes in altering the car and animal's appearance. This distinct specialization in encoding different sets of intra-category content variations is **consistent across various starting images**. We have now added both tables in the Appendix A of the updated manuscript.
>
> | Subject |       Metrics       |    Full Latents    | Unsupervised Latents |
> | :-----: | :-----------------: | :----------------: | :------------------: |
> |   M1    | $R^2$(%) $\uparrow$ | $76.13 (\pm 0.27)$ |  $20.88 (\pm 0.25)$  |
> |   M1    |  RMSE $\downarrow$  | $48.86 (\pm 0.21)$ |  $88.95 (\pm 0.29)$  |
> |   M2    | $R^2$(%) $\uparrow$ | $71.82 (\pm 0.28)$ |  $19.63 (\pm 0.17)$  |
> |   M2    |  RMSE $\downarrow$  | $53.09 (\pm 0.16)$ |    $89.65(±0.31)$    |
>
> *Table 2: Explained neural variance from the unsupervised neural latents of two macaques.*
>
> > In terms of methodology, what is the motivation for the two-step diffusion process (inversion followed by guided synthesis)? Is this intended to constrain image generation to remain on the natural image manifold?
>
> The motivation for conducting the two-step diffusion process is as follows. (1) Starting from a selected reference image, we first perform the DDIM inversion to perturb the high-level visual-semantic details of that image. As our goal is to uncover the high-level semantic features encoded by the neural latent group, we stop the inversion at a middle timestep $t=t'$. (2) We then conduct the diffusion synthesis procedure using the proposed mutual information MI guidance objective from $t=t'$, which directs the generated images to reveal the specific high-level semantic features encoded within the target neural latent group.
>
> It is correct that, after these two steps, the generated images resides on the **natural image manifold**. Notably, our second step of synthesis guides the results toward **specific modes and local regions** of the image manifold. The images residing within these modes and local regions exhibit the encoded visual-semantic features of the target neural latent group.
>
> Refs:
>
> [1] Simple learned weighted sums of inferior temporal neuronal firing rates accurately predict human core object recognition performance. (Majaj et al., 2015)

---

> ### Author Response · Authors · 2025-11-28
>
> > In the diffusion process, does the choice of starting image influence the final outcome?
>
> Yes, that is an insightful observation. We use a specific starting image as the initial reference point for the diffusion process and perturb it using the aforementioned DDIM inversion up to the intermediate timestep $t=t'$. Subsequentely, we apply the proposed mutual-information guidance to denoise and synthesize from this perturbed image to obtain the final output. This procedure allows us to visualize and isolate the specific visual-semantic changes/variations (encoded by the target neural latent group) in the synthesized image compared to the starting image.
>
> This procedure is illustrated in Figure 2 (B) of the revised manuscript. In this MI landscape, the starting images reside on the bottom part, and the MI guidance directs the synthesis procedure toward different final outcomes.
>
> > Finally, the introduction would benefit from clearer framing, as it currently shifts rapidly between topics without defining a specific gap in knowledge or central research question.
>
> Thank you for pointing this out. We have detailedly revised the Introduction section (Line 48-70) of the updated manuscript to clearer frame our research goal. We emphasize that the central research question we aim to solve is to **uncover semantically interpretable neural representations** from electrophysiological recordings in the higher visual cortex.
>
>
>
> We look forward to further discussion, and are glad to answer any questions that may arise.

---

### Official Review · Reviewer_NyBK · 2025-11-01

**Soundness:** 2
**Presentation:** 3
**Contribution:** 2
**Rating:** 4
**Confidence:** 4

**Summary:**

This paper presents a method to visualize how neural populations in the primate brain encode visual information. The approach first processes neural spiking data from macaques through a group-wise disentangled variational autoencoder to identify distinct, low-dimensional latent subspaces. It then employs a diffusion model guided by an objective that maximizes the mutual information between a synthesized image and a target neural subspace, thereby generating images intended to represent the specific visual features encoded by that group of neurons. The resulting images suggest the higher visual cortex organizes information into specialized groups selective for features like object pose, inter-category transformations, and intra-category content details. However, the work's novelty lies more in its combination of existing machine learning techniques than in a fundamental conceptual breakthrough, updating the classic optimal stimulus paradigm with modern tools. Furthermore, the claim of providing direct evidence is questionable, as the visualizations are interpretations generated by a multi-stage modeling pipeline. A significant limitation that tempers the paper's broader conclusions is its reliance on a highly constrained stimulus set consisting of grayscale, segmented objects from only eight categories. This limited visual diversity means the uncovered semantic selectivity is defined by this artificial environment, and claims about the general encoding principles of the higher visual cortex may be overstated, as the findings are not shown to generalize to the complexity of natural vision.

**Strengths:**

The work presents several strengths in its approach to investigating neural representations. It successfully integrates multiple machine learning models, a variational autoencoder and a diffusion model, into a cohesive pipeline for interpreting high-dimensional electrophysiology data.
1. A key contribution is the use of a mutual information-based guidance objective for the diffusion model. The authors support the efficacy of this choice through an ablation study that compares it to a simpler activation-based guidance, showing the MI approach yields more semantically consistent results for complex transformations.
2. The method produces clear and intuitive visualizations of the abstract information encoded in the identified neural subspaces. This translates complex neural activity patterns into understandable images, making the model's findings accessible.
3. The study includes a comparative analysis against several baseline methods for neural guidance. This demonstrates that the proposed approach generates higher-quality and more structurally coherent images, strengthening the claims about its effectiveness.

**Weaknesses:**

The study has several limitations that should be considered when interpreting its conclusions, with the primary concerns relating to the novelty of the approach and the characteristics of the dataset used.
1. The work's main contribution is the novel integration and application of existing machine learning methods to a specific neuroscience problem. The core components: variational autoencoders for disentanglement, diffusion models for generation, and mutual information for guidance. Are all established techniques. As such, the paper represents an advancement in methodology rather than the introduction of a fundamentally new theoretical concept for understanding neural computation.
2. The conclusions are based on neural responses to a highly controlled and simplified visual environment. The dataset consists of grayscale, segmented objects from only eight categories presented on a uniform background. This lack of complexity raises significant questions about the generalizability of the findings. The "semantic" features discovered by the model are defined entirely by the limited variations present in the dataset and may not reflect how the visual cortex represents the much richer and more complex information found in natural scenes, which include color, texture, and cluttered backgrounds.
3. The claim of providing "direct" evidence of neural representation is not strong. The final visualizations are the output of a multi-stage pipeline involving two separate deep learning models. The results are therefore an interpretation of the neural data as filtered through the specific architectures and biases of these models. Different modeling choices could potentially lead to different visual interpretations of the same underlying neural activity.
4. The primary support for the semantic meaning of the latent groups comes from the visual inspection of the synthesized images. While the images are interpretable, the evaluation of their semantic content is largely qualitative and subjective. The study lacks quantitative metrics to formally validate that the generated images accurately capture the intended semantic transformations.

**Questions:**

1. The choice of the VAE architecture, particularly the number of latent groups (four) and the dimensionality of each group (six), is important to the findings. Could you elaborate on the selection process for these hyperparameters? How sensitive are the discovered semantic selectivities to changes in this architecture, for example, if you used more or fewer groups? How would you apply this to natural RGB images?

2. The "latent groups" are a powerful modeling abstraction. Do you have any hypotheses about how these computationally-defined groups might map onto the known anatomical or functional organization of the IT cortex? For instance, could a single latent group correspond to a specific neural sub-population or a known processing stream?

3. Your results show that manipulating a single latent group can produce large, coherent changes in the image, such as transforming a face into a pear. Does this imply that the neural code for these distinct objects is adjacent or connected in this latent space, or is this transformation an artifact of the generative model's ability to interpolate between any two points?

4. Have you considered applying your method to a more diverse dataset to test the robustness of these specific semantic axes? Would you expect to find similar, cleanly separated latent groups, or would you anticipate a more entangled representation?

---

> ### Author Response · Authors · 2025-11-22
>
> Dear Reviewer NyBK,
>
> Thank you for your valuable comments. We would like to provide the following clarifications, which we hope will address most of your concerns and be taken into account in the final evaluation.
>
> > The claim of providing "direct" evidence of neural representation is not strong. The final visualizations are the output of a multi-stage pipeline involving two separate deep learning models. The results are therefore an interpretation of the neural data as filtered through the specific architectures and biases of these models.
>
> This is a critial concern. We first acknowledge that the final visualization images are the output of a two-stage model pipeline. However, we would like to maintain that our proposed MIG-Vis method has its unique power and capability in scientific discovery. To the best of our knowledge, it is the first machine learning tool that can uncover the visual-semantically meaningful neural latent groups from raw higher visual cortex neural recordings.
>
> Crucially, previous works lack the capability to conduct this neural latent group semantic investigation on electrophysiological datasets. Moreover, we have conducted extensive experiments to demonstrate the robustness of the proposed MIG-Vis method.
>
> > The choice of the VAE architecture, particularly the number of latent groups (four) and the dimensionality of each group (six), is important to the findings. Could you elaborate on the selection process for these hyperparameters? How sensitive are the discovered semantic selectivities to changes in this architecture, for example, if you used more or fewer groups?
>
> Thanks for raising this point on hyperparameters. (i) We choose the neural latent group dimensionality $D_g = 6$ because we have a such a number of unique pose labels in the first supervised latent group. We choose the latent groups number $G = 4$ since the total latent dimensionality of $D = 24$ reconstruct the neural data effectively. (ii) We have justified the controllability of the resulting visualizations by varing the latent group number in Fig. 9 of the revised manuscript. We observe that if we reduce the group number to $G = 3$, then the object-specific semantic changes are actually entangled across objects. On the other hand, we increase the group number to $G = 5$, resulting in three object-specific latent groups. We find that one of which (the second row in both figures) is found to be redundant and lacks semantic meaning. Thus we set $G = 4$ across all the objects.
>
> > The "latent groups" are a powerful modeling abstraction. Do you have any hypotheses about how these computationally-defined groups might map onto the known anatomical or functional organization of the IT cortex? For instance, could a single latent group correspond to a specific neural sub-population or a known processing stream?
>
> This is truly a scientific meaningful point. The uncovered neural latent groups can be represented as distinct neural sub-circuits [1] within the higher visual cortex, such as the IT cortex.
>
> > Does the results imply that the neural code for these distinct objects is adjacent or connected in this latent space, or is this transformation an artifact of the generative model's ability to interpolate between any two points?
>
> As the images in the dataset (belonging to various categories) are actually diverse in semantic meanings, our primary aim with MIG-Vis is not to verify the ground-truth neural codings or topography on the manifold. Instead, our proposed framework MIG-Vis provides a computional approach to verify the visual-semantic encodings and transitions on the neural manifold. We note that we are proposing a machine learning tool to help neuroscientisits verify specific semantic encoding hypothesis.
>
> > Have you considered applying your method to a more diverse dataset to test the robustness of these specific semantic axes?
>
> We note that acquiring large-scale and high-quality electrophysiological datasets that meet our specific data format is currently challenging in the neuroscience community. We also note that other works focusing on the scientific discovery with diffusion synthesized images are typically evaluated on a single, high-quality dataset [2].
>
>
>
> Refs:
>
> [1] A Disentangled Low-Rank RNN Framework for Uncovering Neural Connectivity and Dynamics. (Li et al., 2025)
>
> [2] Cortical discovery using large scale generative models. (Luo et al., 2023)

---

> > ### Comment · Reviewer_NyBK · 2025-11-27
> >
> > Thanks! Your response has resolved my questions, I think the work would be a valuable contribution and have increased my score.

---

> > > ### Author Response · Authors · 2025-11-28
> > > **Thank you**
> > >
> > > Dear Reviewer NyBK,
> > >
> > > We appreciate your recognition of the work and the adjustment to your score. Thank you again for your valuable suggestions and comments.

---

### Official Review · Reviewer_X5v2 · 2025-11-03

**Soundness:** 2
**Presentation:** 2
**Contribution:** 3
**Rating:** 4
**Confidence:** 4

**Summary:**

The paper proposes MIG-Vis, a two-stage pipeline to uncover semantically selective latent groups in macaque IT cortex. First, a group-wise (weakly supervised) VAE learns a structured neural latent space. Then, the authors visualize each group’s semantics by editing real images with a deterministic DDIM sampler guided by a surrogate of mutual information (via InfoNCE), so edits stay on-manifold while moving the image toward the target group. Qualitatively, different groups control pose (intra-class), inter-category transitions, and fine intra-class details; ablations suggest stronger, more controllable edits than activation-guided baselines, while disentanglement improves with little loss in neural reconstruction.

**Strengths:**

1. The paper combines a group-wise neural latent space (VAE) with explicit, InfoNCE-guided deterministic DDIM editing, yielding a clean ‘unconditional denoising score + semantic term’ decomposition for controllable edits—going beyond regression guidance and offering a clear, reproducible, and extensible recipe.
2. On macaque IT data, the approach consistently maps different latent groups to pose, inter-category transitions, and fine intra-category details while preserving structure, providing a practical tool for population-level neural interpretation with promising implications for closed-loop neuroscience and interpretable generative modeling.

**Weaknesses:**

1. In Eqs.~(5)--(6) the authors train an InfoNCE scorer and use $-\nabla_y L_{\mathrm{InfoNCE}}$ to approximate $\nabla_y \mathrm{PMI}(z_g,y)$. By standard contrastive–learning results, however, the InfoNCE optimum estimates $\log \frac{p(z\mid y)}{q(z)}$; this coincides with PMI only when the negative–sample distribution satisfies $q(z)\approx p(z)$ and is independent of the current $y$. Therefore, the reliability of the gradient’s direction and magnitude hinges on this assumption. Please (i) specify how negatives are sampled in your implementation and (ii) systematically quantify the impact of different $q(z)$ strategies and batch sizes $B$.
2. The implementation guides sampling with the pointwise mutual information (PMI) gradient for a specific target group $z_g$, yet the main text repeatedly refers to an "MI gradient,'' which can be misread as maximizing the expected mutual information over $p(z\mid y)$. The two have different statistical meanings: PMI means make the current image look more like this group,whereas MI concerns the overall dependence between $Z$ and $Y$. Using "MI'' instead of "PMI" overstates the correspondence between the target and the evidence. Please state explicitly in the main text.
3. The paper fixes the editing start at an intermediate diffusion step $t'$ and employs deterministic DDIM. However, it provides no evidence for why such a $t'$ should generally exist, how $t'$ ought to be selected, or whether the choice is robust across different noise schedules, datasets. Therefore, the observed effects may depend on carefully chosen settings or hyperparameters, rather than the method itself.
4. The number of groups $G$ in the group-wise VAE is a key hyperparameter.  If $G$ is too large, semantic factors tend to fragment across multiple groups;  if $G$ is too small, heterogeneous semantics are merged into the same group.  The authors should provide experiments justifying their choice of $G$ together with metrics such as reconstruction and controllability of the resulting visualizations.
5. While this is a minor issue, it was unclear if the dataset from Majaj et al., 2015 is publicly available but I must specify that MIG-Vis code has been provided on a link. The model is there but I might have missed the data.

**Questions:**

1. In equation (4), the conditional variable $z$ in $\log p_\gamma(y_t \mid z)\,$ is undefined. The authors need replace $z$ by $z_g$ or explicitly define $z$, and ensure consistent notation throughout.
2. The paper states that negative samples are drawn from $q_\phi(z_g \mid \hat{x})$ with $\hat{x}$ unrelated to $y$, but how independence from the current $y$ is guaranteed. The authors need to clarify whether negative samples are drawn cross-image or provide the exact procedure.
3. The quantity in Eq.~(2) is commonly referred to as Total Correlation. Please rename accordingly or explain the difference between your usage and the standard definition.

---

> ### Author Response · Authors · 2025-11-22
>
> Dear Reviewer X5v2,
>
> Thank you for the detailed and constructive comments. We would like to make the following clarifications, which we hope will address most of your concerns and be considered in the final evaluation of our paper.
>
> > Please (i) specify how negatives are sampled in your InfoNCE scorer implementation and (ii) systematically quantify the impact of different $q(z)$ strategies and batch sizes .
>
> (i) We confirm that we approximate the ground-truth overall latent distribution $p(\mathbf{z})$ using $q(\mathbf{z})$. In our InfoNCE scorer [1] implementation, given a batch size $B$ and a positive latent sample $\mathbf{z}^{(i)}$, we sample $B-1$ negative latents from the neural latents {$\{ \mathbf{z}^{(j)} \}_{j \neq i}$} that do not correspond to the original image. (ii) We compare the experimental results among $B = 8$, $B = 16$ and $B = 32$. The results show that our batch size setting of $B=16$ generally yields more stable and clear semantic transitions across various starting images and guidance scales compared to $B=8$.
>
> > Using "MI'' instead of "PMI" overstates the correspondence between the target and the evidence. Please state explicitly in the main text.
>
> Thank you for raising this point regarding terminology. Our method is maximizing the **group-wise MI ("GMI")** between the neural latent group and the synthesized image. We clarified this throughout the revised manuscript.
>
> > The paper provides no evidence for why such an intermediate diffusion step $t'$ should generally exist, or whether the choice is robust across different noise schedules.
>
> We agree that the intermediate diffusion step $t'$ to start synthesis is a critial hyper-parameter. For a total of $t=150$ diffusion timesteps, we set $t'=135$ among all the latent groups for a fair comparison. This setting is based on the properties of diffusion models that the first several denoising steps (closer to $t=150$) primarily recovers the low-level, basic visual features. While our diffusion guidance procedure aims to uncover the high-level visual-semantic features of the target neural latent group, which exists in the later stages of diffusion denoising.
>
> > The authors should provide experiments justifying their choice of $G$ together with metrics such as reconstruction and controllability of the resulting visualizations.
>
> That's a critical point, and we have justified them in the added experiments in Figure 9 of the revised manuscript. We observe that if we reduce the group number to $G = 3$, then the object-specific semantic changes are actually entangled across objects. On the other hand, we increase the group number to $G = 5$, resulting in three object-specific latent groups. We find that one of which (the second row in both figures) is found to be redundant and lacks semantic meaning. Thus we set $G = 4$ across all the objects.
>
> > While this is a minor issue, it was unclear if the dataset from Majaj et al., 2015 is publicly available
>
> Yes. This dataset [2] on primate Inferior Temporal (IT) cortex neural recordings is publicly available.
>
> > In equation (4), the conditional variable $z$ is undefined. The authors need replace $z$ or explicitly define $z$, and ensure consistent notation throughout.
>
> Thank you for pointing this out. We have corrected this minor issue by replacing $\mathbf{z}$ with $\mathbf{z}_g$ in the updated manuscript.
>
> > The paper states that negative samples are drawn from $q_\phi(z_g\mid\hat{x})$ with $\hat{x}$ unrelated to $y$, but how independence from the current $y$ is guaranteed.
>
> We appreciate your request for explicit clarity. As stated in our response to the first point, the independence is guaranteed by the in-batch sampling strategy. In our InfoNCE scorer implementation, we sample $B-1$ data points from the neural latents set {$\{ \mathbf{z}^{(j)} \}_{j \neq i}$} that does not correspond to the original image $\mathbf{y}^{(i)}$.
>
> > The quantity in Eq.~(2) is commonly referred to as Total Correlation. Please rename accordingly or explain the difference between your usage and the standard definition.
>
> We appreciate the clarification regarding the terminology. We note that the term in Eq. (2) focuses on the group-wise factorized distributions, in which we enforce the **disentanglement between mult-variate latent groups**. Thus we denote the related loss term as *Partial Correlation* (PC) [3]. By comparison, the standard Total Correlation (TC) encourages disentanglement between *individual* neural latent dimensions.
>
>
>
> We look forward to further discussion, and are glad to answer any questions that may arise.
>
> Refs:
>
> [1] Representation Learning with Contrastive Predictive Coding. (Aaron et al., 2018)
>
> [2] Simple learned weighted sums of inferior temporal neuronal firing rates accurately predict human core object recognition performance. (Majaj et al., 2015)
>
> [3] A Revisit of Total Correlation in Disentangled Variational Auto-Encoder with Partial Disentanglement (Li et al., 2025)

---

### Meta-Review · Area_Chair_s2UH · 2026-01-05

**Summary:**

Concerns centered on the mathematical rigor and motivation (X5v2, bCGZ), hyperparameter sensitivity (X5v2, NyBK), the reliance on limited datasets (NyBK, 4PLU), the lack of conceptual novelty (NyBK, 4PLU), the lack of direct evidence (NyBK), the subjectivity analyses (NyBK, bCGZ), the limited explained variance (bCGZ, 4PLU).

**Reviewer Concerns:**

The authors adequately addressed concerns regarding the hyperparameter sensitivity, limited explained variance.

The concerns about the reliance on limited datasets, subjectivity of the analyses, lack of conceptual novelty and the lack of direct evidence remain.

**Reviewer Scores:**

I think that Reviewer X5v2 would have increased their score from 4 to 6 since many of their concerns were addressed.

I think that Reviewer NyBK would have increased their score based on the author feedback from 4 to 6.

I think that some of Reviewer bCGZ's concerns were addressed and they would have increased their score from 2 to 4.

I think that Reviewer 4PLU would have maintained the same score (6).

---

### Decision · Program_Chairs · 2026-01-26

Accept (Poster)